# Consolidation of Additive Manufactured Continuous Carbon Fiber Reinforced Polyamide 12 Composites and the Development of Process-Related Numerical Simulation Methods

**DOI:** 10.3390/polym14163429

**Published:** 2022-08-22

**Authors:** Stefan Grieder, Igor Zhilyaev, Marco Küng, Christian Brauner, Michael Akermann, Jonas Bosshard, Petra Inderkum, João Francisco, Yannick Willemin, Martin Eichenhofer

**Affiliations:** 1Institute of Polymer Engineering, FHNW University of Applied Sciences and Arts Northwestern Switzerland, Klosterzelgstrasse 2, 5210 Windisch, Switzerland; 29T Labs, Badenerstrasse 790, 8048 Zürich, Switzerland

**Keywords:** additive manufacturing, fiber-reinforced composites, additive fusion technology, consolidation, composite additive fusion simulation

## Abstract

Additive manufacturing of high-performance polymers—such as PA12, PPS, PEEK, and PEKK—combined with industrial-grade carbon fibers with a high fiber volume ratio of up to 60% allows a weight reduction of over 40% compared to classic metal construction. Typically, these 3D-printed composites have a porosity of 10–30% depending on the material and the printing process parameters, which significantly reduces the quality of the part. Therefore, the additive manufacturing of load-bearing structural applications requires a proper consolidation after the printing process—the so-called ‘additive fusion technology’—allowing close to zero void content in the consolidated part. By means of the upfront digital modeling of the consolidation process, a highly optimized composite component can be produced while decreasing the number of expensive prototyping iterations. In this study, advanced numerical methods are presented to describe the consolidation process of additive manufactured continuous carbon fiber reinforced composite parts based on the polyamide 12 (PA12) matrix. The simulation of the additive fusion step/consolidation provides immediate accuracy in determining the final degree of crystallization, process-induced deformation and residual stresses, final engineering constants, as well as porosity. The developed simulation workflow is demonstrated and validated with experimental data from consolidation tests on the final porosity, thickness, and fiber–volume ratio.

## 1. Introduction

Additive manufacturing is one key driver which combines digital automated manufacturing, flexibility, and on-demand production. Additionally, it enables lightweight design as well as near net shape manufacturing. For polymer materials, several methods exist, such as fused filament fabrication (FFF), stereolithography (SLA), and selective laser melting (SLM) [1]. The disadvantage of these methods is that most of the time, the pure polymer produced is limited to non-load-bearing applications. Using classical FFF methods in combination with endless fibers, the challenge is to manufacture primary load-carrying structures. Similar to classical composite manufacturing methods—such as automated fiber placement (AFP), automated tape laying (ATL), or tailor fiber placement (TFP)—continuous fiber additive manufacturing uses pre-impregnated fibers as well. As a matrix material, high-performance thermoplastic polymers such as PA12, PPS, PEEK, or PEKK are mainly used. The reinforcement material was either carbon or glass fibers. The nature of the composite reinforcement is determined by the length of the fiber, i.e., short-fiber reinforcement (SFR) or continuous-fiber reinforcement (CFR) [2]. SFR is easy to integrate because it can be added to the polymer filament. However, if there is a variation in the filament diameter or non-uniformity of the fiber volume fraction or size of the fibers (typically between 50 and 80 µm), nozzle clogging can occur. On the other hand, CFR does not require an extrusion process, and the printed parts are more uniform, providing higher mechanical performance values. The disadvantage is that special modifications of the print head are required [2]. Therefore, different companies—such as Markforged (Watertown, MA, USA), Anisoprint (Monnerich, Luxembourg), Orbital Composites (San Jose, CA, USA), or 9T Labs (Altstetten, Zürich)—have addressed this challenge and have developed 3D printers for CFR [3,4,5].

Blok et al. addressed the issue of porosity during FFF manufacturing for SFR and CFR [3]. They stated: “A disadvantage of the continuous fiber printer, however, is limited control over the placement of the fiber and the creation of voids when printing more complex shapes”. However, the porosity problem is a drawback of any additive manufacturing method. If the print speed is increased, the porosity increases, leading to lower mechanical performance values, mainly interfacial shear strength [5,6,7,8,9,10]. Hence, the requirements for a load-bearing part are no longer fulfilled. Different approaches can be used to tackle this problem, for example by heat treatment during printing [11], annealing, or compaction during post processing [12,13]. There are two developments related to compaction. On the one hand, integrating a compaction unit into the printing process by additional rollers is the state of the art in automated tape laying. On the other hand, it is possible to use a consolidation process using a press and a solid mold.

During consolidation, many phenomena—such as bulk compaction, intimate contact between adjacent plies, interlaminar adhesion, fiber deformation and movement, as well as molecular diffusion—occur simultaneously [7]. These phenomena have complex interactions and are influenced by the process parameters time, temperature, and pressure. In order to investigate the consolidation process, several levels can be classified, such as the macro, micro, or molecular level. On the macro level, the relationship between mechanical performance (strength/void content, etc.) and the main process parameters of pressure, time, and temperature can be studied. On the micro level, the scale of fiber diameter and/or tow diameter along with the wetting and flow behavior can be analyzed. For instance, how the matrix fills empty spaces between the fibers can be assessed. At the molecular level, the intimate contact between two adjacent layers and the diffusion of molecular chains across the ply interface can be observed [8]. This type of porosity is different compared to standard composite processes in which the polymer is pressed into a dry filament. In the use case considered in this study, the porosity is in between the printed filaments and therefore a different understanding is needed, as well special consolidation methods respective dedicated numerical methods.

The simulation of the consolidation topic is not a trivial task, which involves coupled multiphysical modeling of the various physical phenomena. Complexity and novelty of the problem leads to the following literature gaps in the model development methodology:Definition of the composite material stiffness in the molten and transitional from solid to molten states (dependency of the stiffness on the crystallization degree) is not presented well in the literature due to the problematic measurements of the material properties in these material states. Therefore, empirical relations are suggested, which need to be adapted for a particular case.Composite stiffness, thermal expansion, and other engineering constants dependent upon the porosity and additive manufactured structure are presented in literature mostly for the solid material state; while in the molten and transitional state, such dependency is not possible to measure due to the permanent evolution of the void content during the measurement due to the forces and temperature applied.Simulation of the air evacuation through the void channels and its dissolution into the surrounding matrix depending on the temperature and pressure applied is not fully covered in literature.Fast void collapse mechanisms during the composite’s consolidation are not yet covered by the literature in relation to the composites consolidation.

The company 9T Labs has developed a unique set of methods to design parts using the Fibrify^®^ Design Suite to print parts with the “Build Module” and to consolidate the part using the so-called “Fusion Module” (see Figure 1).

The analysis of short beam tests following ASTM D2344 showed great improvement in the interlaminar shear behavior due to this fusion process (see Figure 2). It is stated at this point that the interlaminar strength is not equal to the short beam strength; however, it is a qualitative measure to determine the change in interlaminar bonding.

Optical microscopy was performed for different test specimens fused at various temperatures. The tests confirmed the assumption that the fusion process is not independent of the fusion temperature. The fusion temperature influences the flowability/viscosity of the fiber-reinforced material as well as the pressure created by the difference in thermal expansion. Examples of the composite consolidation at various temperatures are presented in Figure 3. Studies have not yet included time effects, which are also expected to be present.

The following study presents a novel method to represent this fusion step, the consolidation, to analyze the dependence of the final porosity and, consequently, the effects of mechanical properties on the process conditions. The main objectives of this study are the following:Composite material model development for PA12-CF.Development of the semi-empirical dependencies of engineering mechanical properties on porosity and crystallization.Simplified approach for porosity development and integration into the commercial finite-element software Ansys via a user-defined material subroutine.Development of the coupled thermo-mechanical approach, including the mechanisms of the porosity evaluation during the process depending on the pressure and temperature applied for the simulation of the additive fusion via a user-defined material subroutine.Validation of the proposed numerical approach on the basis of the simple geometry.

The method is implemented by a sequential thermo-mechanical coupled transient implicit analysis in Ansys R2022 (Canonsburg, PA, USA) based on user subroutines. In the thermal part, the local temperature distribution is calculated, considering temperature-dependent heat capacity, density, and thermal conductivity. Using these local temperatures inside the part, the phase transition behavior of the polymer from solid to molten and back to solid is modeled considering a crystallization approach, namely a modified Nakamura model. In mechanical parts, all engineering properties—such as the Young’s modulus and thermal shrinkage coefficients—are dependent on temperature, fiber volume content, crystallization, and porosity. This numerical method is validated by experimental consolidation tests and measurements of the porosity by different methods such as computer tomography and micro section analysis.

## 2. Materials, Methods, and Characterization

In this section, the materials and methods are described as well as the measurements including results, which are required for the development of the numerical method to understand the consolidation behavior. The developed mathematical model is presented in the following section.

### 2.1. Material and Sample Preparation

The materials used in this study were a carbon fiber-reinforced material with a polyamide 12 matrix (PA12-CF) provided by 9T Labs [14] and an unfilled polyamide 12 (PA12). All the properties of the neat polymer, the high tensile fibers used, and the final composite PA12-CF are listed in Table 1.

The specimens which have been used in the study are printed using the Build Module developed by 9T Labs. The Build Module is an FFF printer that can place pre-impregnated, continuous fibers and pure plastic filaments in a heated print chamber. The continuous fiber is cut during the printing process whenever a fiber path is completed. At the starting point of the fiber path, an anchoring process takes place to ensure that the fiber is placed well and adheres properly to the previous layer. The printing parameters that were used can be found in Table 2.

### 2.2. Measurement Methods and Results

To start the development of the numerical method and to obtain validation data, different experiments were carried out. First, the temperature dependence of stiffness was measured by means of dynamic mechanical analysis (DMA) combined with a rheometer test. Second, a unique consolidation setup was used in a universal tensile test machine, allowing pressure application during the additive fusion process. Third, X-ray computer tomography was applied to measure the porosity percentage with respect to different consolidation parameters such as temperature and pressure.

#### 2.2.1. Measurement of Temperature-Dependent Stiffness

Neat PA12 polymer specimens were characterized by three-point bending tests in a dynamic mechanical analysis (Q800 DMA, TA instruments, New Castle, DE, USA). The dimensions of the sample used were 32/10/5 mm (length/width/thickness). The measurement was performed at a frequency of 0.1 Hz and with a temperature ramp of 3 °C/min starting at 25 °C and rising to 165 °C.

To measure above the melting point of around 185 °C, plate-to-plate rheometer measurements were performed using a plate-to-plate setup with an MCR 300 rheometer from Physica. A plate diameter of 25 mm with a gap of 1 mm was used. A temperature ramp of 3 °C/min starting at 180 °C and rising to 265 °C and an excitation of 0.5 rad/s were applied.

A polynomial curve fitting with a 10th-degree dependency for the DMA results and a 5th-degree dependency for rheometer results was used to represent the obtained data as a function of temperature. Measured data and corresponding fits are shown in Figure 4.

#### 2.2.2. Measurement of the Consolidation Behavior

A square-shaped (32 × 32 × 4 mm) consolidation test setup was designed in combination with a universal testing machine (K100, Zwick Roell universal testing machine, Ulm, Germany) to apply the pressure precisely and to measure the displacement. The temperature-dependent bulk modulus and the interaction between the applied pressure and the porosity were determined with a test program using temperatures below (175 °C) and above the melting temperature (185 °C) and with pressure levels from 1 to 2 MPa.

The lower part of the setup consisted of a heated frame bolted to a water-coolable base plate. This plate was then positioned on an insulating plate which itself was put on a heavy steel plate to reduce compliance. The upper setup consisted of a water-coolable stamp bolted to the upper insulation plate along with a heating plate and mounted on the load cell of the testing machine via a water-cooled adapter (see Figure 5).

The specimen used was a rectangular-shaped, unidirectional PA12-CF specimen (32 mm × 32mm × 4 mm). The specimens were cut from longer PA12-CF bars using a water-cooled band saw and then dried at 80 °C in a vacuum oven for at least 4 h before consolidation in the experimental setup.

The specimens were measured before and after the consolidation test with a caliper gauge in width and length and with a micrometer in thickness at five different positions. In addition, the specimens were weighed before and after the consolidation experiment to determine the amount of outflow during the experiment.

The samples were heated up to melting temperature without applying any pressure. After reaching the goal temperature, the samples were pressed at a rate of 0.5 mm/min until the objective pressure was obtained. The pressure afterwards was maintained for 10 min to achieve a uniform process condition within the sample. After 10 min, the heating was switched off and the water cooling of the stamp and the base plate were switched on and the whole setup was cooled down to room temperature. After reaching room temperature, the sample was demolded and the dimensions and weight were measured. Process description is presented in Figure 6.

The compaction of the individual specimens was derived from the consolidation experiment setup. Compaction was calculated as
(1)$$Compaction=t_{init}-t_{final}$$
where $t_{init}$ is the average initial thickness measured with a micrometer and $t_{final}$ is the average final thickness measured with a micrometer. Based on this, the porosity ϕ was determined by
(2)$$\phi =1-\frac{\rho }{\rho_{0}}$$
(3)$$\rho =\frac{m}{V}$$
where ρ is the density of the sample including porosity, $\rho_{0}$ is the consolidated material specific density, m is the mass of the sample, and V is the volume of the sample. The results of the proposed method for the initial and porosity calculation are presented in Table 3.

The initial porosity varies from sample to sample depending on the properties during manufacturing. Moreover, even with the same printing parameters, the initial porosity has about 3% variation from the mean value.

From the consolidation experiment, the bulk modulus of the molten polymer can be derived using the stress and strain data. To exclude the compliance of the machine and the consolidation experiment setup, the bulk modulus $E_{m,b}$ was calculated at maximum stress during the pressure application.
(4)$$E_{m,b}=\frac{\sigma_{max}-\sigma_{half}}{\epsilon_{\sigma_{max}}-\epsilon_{\sigma_{half}}}$$
where $\sigma_{max}$ is the maximum stress during pressure application, $\sigma_{half}$ is half of the maximum stress during pressure application, $\epsilon_{\sigma_{max}}$ is the elongation at $\sigma_{max}$ and $\epsilon_{\sigma_{half}}$ is the elongation at $\sigma_{half}$. Formula (4) is a classical definition of the stiffness, while the value of $\sigma_{half}$ is defined by the authors based on the behavior of the stress–strain curves.

As visible in Figure 7 the temperature has a large impact on the bulk modulus and as well on the final porosity. Average bulk modulus for samples A, D, and E evaluated according to (4) is 37.8 MPa. However, the deviation from the mean value is about 6 MPa, while the method itself is highly dependent on the definition of $\sigma_{half}$. Samples B and C were excluded from the analysis due to the low process temperature and pressure.

#### 2.2.3. Advanced Measurement of Porosity

Computed tomography (CT) was used to determine initial and final porosity. CT scans were performed with X-ray Phoenix V|tome|x M (Boston, GE, USA) with a voxel size of 25–31 µm, an accelerating voltage of 120–150 kV, a tube current of 100–500 µA, and a power of 20–75 W. The evaluation was performed with the post processing software Volume Graphis (VG, Heidelberg, Germany). In Figure 8, an unconsolidated sample is shown. The results of the CT analysis for the consolidated sample A are shown in Figure 9.

Dependent on the shape of pores, it may be difficult to derive a valid value for the porosity content. In the considered case, the pore is not a sphere, but more tubular and mostly open for the unconsolidated samples. The general algorithm to obtain these data does not capture open pores, and tubular-shaped pores are not covered properly. To overcome this problem, an approach with a sub-region of interest (ROI) was used and the range of various CT analysis settings were tested to provide an interval of confidence. However, a reliable interval of the porosity percentage was obtained only for consolidated samples A and C (0.05–0.15%), while porosity values vary significantly depending on the CT analysis settings for the unconsolidated (10–25%) and partially consolidated (sample B: 0.11–7.23%) samples. Therefore, porosity values evaluated on the basis of the density-based approach presented above (see Table 3) were implemented for the numerical simulation validation, since they are within the confidence interval and more accurate measurement was not available.

## 3. Development of Constitutive Equations

In the following section, the developed model (porosity approach, stiffness’ dependency on porosity, and temperature and homogenization methods for the engineering properties of the composite) is presented. The presented mathematical model is then integrated into the finite-element approach to simulate the consolidation test from the previous section and to validate the proposed numerical approach.

### 3.1. Porosity Model

Porosity ϕ is presented in the model as a dimensionless variable, which takes a value between 0 and 1, where 0 means that the material is fully consolidated and 1 means that the whole considered finite element’s volume is empty. The initial porosity value depends on the material and the printing setup (see Table 3).

In the present study, the following mechanisms of the porosity evolution during the consolidation process were considered [15]:

Porosity reduction due to the void regions’ compression by the high external pressure (ideal gas law)Porosity reduction due to the molten resin flow into the void regions from the neighboring regions (Darcy squeeze flow)Porosity reduction due to the trapped air dissolution into the resin (Henry’s law)

Hydrostatic pressure $p_{h}$ is evaluated according to Barari et al. [15]
(5)$$p_{h}=\{\begin{array}{c} P_{0}\cdot \frac{\phi_{0}}{\phi }\frac{T}{T_{0}}\left(1-V_{d}\right),\phi >\phi_{min}\\ 0,\;\phi \leq \phi_{min} \end{array}$$
where $P_{0}$ is the initial (atmospheric) pressure, $T_{0}$ is the initial (room) temperature, $\phi_{0}$ is the initial porosity, $\phi_{min}=0.001$ is the minimal considered porosity, and $V_{d}$ is the relative volume of dissolved air.

The volume of the dissolved air is a dimensionless variable, which takes values between 0 and 1, where 0 means that air occupies 100% of its initial volume and 1 means that air is completely dissolved in the surrounding resin. It is assumed that air traps have spherical shapes and therefore volume can be defined as
(6)$$V_{d}=1-{R_{b}}^{3}$$

$R_{b}$ is a dimensionless variable which represents the air bubble radius and takes values between 0 and 1, where 0 means that the air bubble is completely dissolved into the resin and 1 means that bubble radius is 100% of its initial value. Therefore, $R_{0}=1$. Relation (6) is a semi-analytical approach based on the dimensionless definition of the dissolved volume and bubble radius. 

The relative bubble radius is evaluated as [16,17]
(7)$$R_{b}={\left(R_{0}-D\cdot \left(p_{h}-p_{h,l0}\right)\cdot t\right)}^{0.5}\;$$
where D is a dimensionless diffusion model parameter, and $p_{h,l0}$ is the hydrostatic pressure when melting occurs.

Squeeze pressure $p_{sq}$ is only considered as orthotropic during the liquid state and is evaluated according to Barari et al. [15]
(8)$$p_{sq,i}=\{\begin{array}{c} \frac{\eta \left(T\right)}{12\cdot K_{i}}\cdot {\Lambda_{i}}^{2}\cdot \frac{\partial \varepsilon_{m,i}}{\partial t},\phi >\phi_{min}\\ 0,\;\phi \leq \phi_{min}\;or\;i=1 \end{array},i=\left\{2,3\right\}$$
where η(T) is the molten matrix viscosity, and $\Lambda_{i}$ is the directional distance to the closest void (defined by the printing setup). Due to the composite manufacturing method, voids are represented as channels following the printing direction. Therefore, the distance to the closest void in the printing direction $\Lambda_{1}$ is assumed to be 0, meaning that there is no squeeze flow in printing direction. $\varepsilon_{m1},\varepsilon_{m2},\varepsilon_{m3}$ are strains of the matrix material inside the fiber filament, evaluated according to Schürmann [18]
(9)$$\varepsilon_{m,i}=\frac{\varepsilon_{i}}{l_{m}/l_{0}+E_{m,r}/E_{f,i}\cdot \left(1-l_{m}/l_{0}\right)},\;i=\left\{1,2,3\right\}$$
where $\varepsilon_{1},\varepsilon_{2},\varepsilon_{3}$ are the strains in the longitudinal, transverse, and thickness direction, respectively; $E_{m,r}$ is the matrix elastic modulus; $E_{f,i},i=\left\{1,2,3\right\}$ are the elastic moduli of the fiber in the longitudinal, transverse, and thickness direction ($E_{f,2}=E_{f,3}$); and the $l_{m}/l_{0}$ ratio is defined as [18]
(10)$$l_{m}/l_{0}=1-{\left(\frac{4}{\pi }\varphi \right)}^{0.5}$$

Here, φ is the fiber volume ratio. By simplifying the interlayer region into a rectangular duct, the permeability in the transverse and thickness direction is defined as [19]
(11)$$K_{2}=\frac{a^{2}}{12}\left[1-\frac{192\cdot a}{w\cdot \pi^{5}}\sum_{i=1,3,5,\ldots }^{9}\frac{\tanh h\left(\frac{i\cdot \pi \cdot w}{2\cdot a}\right)}{i^{5}}\right]$$
(12)$$K_{3}=\frac{w^{2}}{12}\left[1-\frac{192\cdot w}{a\cdot \pi^{5}}\sum_{i=1,3,5,\ldots }^{9}\frac{\tan h\left(\frac{i\cdot \pi \cdot a}{2\cdot w}\right)}{i^{5}}\right]$$
where *a* and *w* are the height and width of the rectangular duct, respectively, evaluated according to
(13)$$w=w_{0}\cdot \left(1+\varepsilon_{2}\right)$$
(14)$$a=a_{0}\cdot \left(1+\varepsilon_{3}\right)$$
where $w_{0}$ and $a_{0}$ are the printed filament width and layer height, respectively (see Table 2). Porosity is evaluated as a function of the bulk strain $\varepsilon_{bulk}$ as [15]
(15)$$\phi =\{\begin{array}{c} \phi_{0}+\varepsilon_{bulk},\phi >\phi_{min}\\ 0,\;\phi \leq \phi_{min}\; \end{array}$$
(16)$$\varepsilon_{bulk}=\left(1+\varepsilon_{1}\right)\cdot \left(1+\varepsilon_{2}\right)\cdot \left(1+\varepsilon_{3}\right)-1$$
where $\phi_{0}$ is the initial porosity, which is evaluated according to our measured initial porosity of the unconsolidated composite part. Initial porosities of the considered samples are defined with CT analyses and presented in Table 4.

### 3.2. Stiffness Dependency on Porosity and Temperature

The matrix elastic modulus $E_{m}$ depends on the temperature, crystallization degree, and porosity *ϕ* as [19,20]
(17)$$E_{m}=E_{m,s}\cdot {\left(\theta \right)}^{K}\cdot {\left(1-\phi_{t}\right)}^{S}+E_{m,b}\cdot {\left(1-\theta \right)}^{K}\cdot {\left(1-\phi_{t}\right)}^{M}$$
where $E_{m,b}$ is the molten polymer bulk modulus measured in the pressure evaluation study and $E_{m,s}\left(T\right)$ is the temperature-dependent elastic modulus in the solid state; K=1.85, S=7.5, and M=75 are model parameters; and $\phi_{t}$ is “true” porosity, which considers the dissolved volume influence as
(18)$$\phi_{t}=\phi \cdot \left(1-V_{d}\right)$$

The minimal possible value of the elastic modulus $E_{m}$ is limited by the measured in the rheology experiment temperature-dependent stiffness function (see Figure 4). Formula (18) is a relation derived by the authors based on the proposed dimensionless approach for both porosity and dissolved volume.

The change in the material state from solid to liquid and back is described by the Richards function for melting [21], and the Nakamura equations for the crystallization [22,23,24] fit to the DSC data. The details regarding the description of melting and crystallization behavior are outside the scope of this paper and will be published by the authors in a future paper, while in general the approach is presented by the authors in [25,26].

### 3.3. Homogenization

In the previous section, a porosity model was developed and the stiffness’ dependence of neat PA12 polymer on porosity and temperature was described. In the next section, homogenization methods are derived to transfer the findings to the composite level.

The fiber volume ratio φ linearly depends on the porosity using the assumption
(19)$$\varphi =\varphi_{0}\left(1-\phi \right)$$
where $\varphi_{0}$ is the measured fiber volume ratio, which corresponds to the fully consolidated PA12-CF. Relation (19) is an empirical formula derived by the authors based on the structure of the 3D-printed composite and dimensionless approach for both fiber volume ratio and porosity.

The transversal isotropic mechanical properties are evaluated according to a homogenization approach from Halpin-Tsai [27], which depend on the temperature and fiber volume ratio. The elastic modulus in the fiber direction $E_{1}$ is defined as [18]
(20)$$E_{1}=E_{f,1}\varphi +E_{m}\left(1-\varphi \right)$$

The elastic moduli in the transverse and thickness direction, respectively, are defined as [27]
(21)$$E_{2}=E_{3}=E_{m}\frac{1+\zeta_{1}\cdot \varphi \left(E_{f,2}-E_{m}\right)/\left(E_{f,2}+\zeta_{1}\cdot E_{m}\right)}{1-\left(E_{f,2}-E_{m}\right)/\left(E_{f,2}+\zeta_{1}\cdot E_{m}\right)\varphi }$$

Here, $\zeta_{1}=3.3$ is the model parameter fitted to the experimental data for PA12-CF to provide the modulus value close to the measured value.

Poisson’s ratio is defined according to [18]
(22)$$v_{12}=v_{13}=v_{23}=v_{f}\varphi +v_{m}\left(1-\varphi \right)$$

Shear moduli are defined as follows [18,27]: (23)$$G_{m}=\frac{E_{m}}{2\left(1+v_{m}\right)}$$
(24)$$G_{12}=G_{13}=G_{m}\frac{1+\zeta_{2}\cdot \varphi \cdot \frac{G_{f,12}-G_{m}}{G_{f,12}+\zeta_{2}\cdot G_{m}}\;}{1-\varphi \cdot \frac{G_{f,12}-G_{m}}{G_{f,12}+\zeta_{2}\cdot G_{m}}\;}$$
(25)$$G_{23}=\frac{E_{2}}{2\left(1+v_{23}\right)}$$

Here, $\zeta_{2}=1.675\cdot$ is the model parameter fitted to the experimental data for PA12-CF to provide a modulus value close to the measured value.

The presented mixing rules (19)–(25) hold for the PA12-CF filament in the solid state, while in the molten state, the material is considered to be isotropic, with material properties corresponding to the pure PA12.

The developed model considers the mixing rules for the orthotropic CTE and the crystallization shrinkage coefficients, which are based on [18,20,21]. The approach in general was presented by the authors in [25,26]. However, this topic is outside of the scope of this paper and will be discussed in detail in an upcoming publication.

## 4. Model Application and Validation

In this section, the application of the developed model into the Ansys environment is described. Results of the thermal and mechanical solutions are presented—including crystallization, process-induced compaction, residual stresses, porosity, and squeeze pressure. The model is validated based on measured porosity and final specimen compaction (thickness change).

### 4.1. Model Application to Finite-Element Method

The developed approach was implemented into Ansys using user subroutines. A sequential coupled thermal mechanical analysis was used. In the thermal model, the full consolidation unit was modeled using 232,041 tetrahedral elements. The temperature shown in Figure 10 was used in the thermal analysis. In the mechanical problem only, the composite part was modeled using 48,588 hexahedral elements. The local temperatures were transferred from the thermal solution to the mechanical problem and the pressure related to the different consolidation trails was applied. 

The thermal model considers three phases of the consolidation process for the whole mold and composite part inside, namely heating, melting, and cooling. Heating in the experiment is controlled by the PID controllers in the mold. The simulation provides a temperature regime corresponding to the experimental setup by manually setting the heat flow on the heating cartridges leading to the corresponding simulated temperature in control points. The boundary conditions for the thermal model are the convection of all the outer mold surfaces with the surrounding air as well as the radiosity of the outer walls. The initial temperature is equal to room temperature (22 °C).

The mechanical model resolves the problem for the composite part only. In the presented approach, a sequentially coupled thermal-stress analysis is performed, in which the temperature field does not depend on the stress field. The boundary conditions are shown in Figure 11. The release from the mold is simulated after 3750 s by removing all the boundary constraints and forces applied and providing a three-point fixation, which allows the specimen to deform freely.

The simulation time for the considered problem is 42 min for the thermal problem and 90 min for the mechanical problem using 4 Intel Xeon Gold 2.7 GHz processors and 64 Gb RAM on a 64-bit operating system. The increased solution time for the mechanical problem despite the lower number of finite elements is explained by the implementation of the user-defined mechanical model subroutine, which executes multiple additional arithmetical operations in every iteration step.

### 4.2. Results of the Thermal Analysis

In Figure 12, the average temperature in the blockply domain is shown, including its variation. The maximum temperature is around 190 °C (samples A, C, D, E) and around 180 °C (sample B). The maximum temperature variation of around 20 °C arises during cooling, the maximum heating rate is 16 °C/min and the maximum cooling rate is 24 °C/min.

In Figure 13, the relative degree of crystallization is shown. Due to the consolidation temperature (175 °C) for sample B, no complete melting happens. Relative crystallinity of about 18% is reached on average for sample B (crystallization range for sample B is 10–24%). For other samples, the relative crystallization reaches 0, corresponding to the complete molten state.

As mentioned in the model description in Section 3, the relative crystallization is used as a state variable to integrate the phase changes into the model and to provide the stiffness dependence on the material state. Despite the fact that composites are usually consolidated within the temperature above the melting point [7,8,9], sample B consolidated with the lower temperature and, therefore, not reaching zero relative crystallization is a point of interest for the model feasibility check.

### 4.3. Results of the Porosity Analysis

Figure 14 shows the change in the porosity during the consolidation process. Depending on the single measurements of the initial porosity taken from Table 3, the starting point of the porosity is different. At around 2300 s, a pressure of 2 MPa for samples A, B, D, E, and a pressure of 1 MPa for sample C are applied.

In comparison to the measured final porosity, all results are below 0.1% for the samples which are consolidated above the melting temperature (see Table 4). Sample B, which is consolidated 5 °C below melting temperature, shows a final porosity of 4.96% and the value obtained with the simulation is 3.7%. The reason for this deviation might be an imperfection of the developed simplified model, an inaccuracy of the CT measurements, poor model tuning, or other factors. The presented results correspond to the ones presented in literature, where a successful consolidation leading to the close to zero final porosity might be reached only within the application of the temperature above melting point [7,8,9].

In addition to the porosity, the hydrostatic pressure in the void and the relative bubble radius can be analyzed for the different process conditions (Figure 15 and Figure 16). The problem of the bubble collapse is non-trivial due to the rapid change of the local material properties and usually not covered in literature devoted to the simulation of composites consolidation [15,16]. The simulated hydrostatic pressure for the fully consolidated samples shows significant and rapid growth (above 50 MPa) during the pressure application and bubble collapse around 2300 s, which leads to the convergence problem and, therefore, it was not considered in the constitutive equations. Figure 15 and Figure 16—as well as the following Figures 19, 21 and 22—show results only for two samples A and B since the solution for the presented variables is similar to the sample A. For example, the solution for the hydrostatic pressure for samples C, D, and E provides only slightly different values of the maximum void pressure during the external pressure application following the same trend.

Figure 15 and Figure 16 show that void collapse happens within a short time for the sample A, causing significant and rapid increase in the internal void pressure. Advanced models of bubble collapse in a viscoelastic medium might be implemented for better accuracy of bubble collapse simulation.

In contrast to the hydrostatic pressure, the squeeze pressure is low and can be analyzed in a certain direction, for instance transverse to the fiber direction (see Figure 17).

In the considered application case, the sample is a simple rectangular plate and, therefore, the hydrostatic and squeeze pressure are uniformly distributed in the domain. This situation will change if the model is transferred to a more complex case with non-uniform temperature and pressure application, more advanced fiber filament mapping or curved geometries with varied thickness will be required. Related to the resulting squeeze pressures, a filament movement analysis could be performed on the basis of the directional squeeze pressures distribution.

### 4.4. Results of the Mechanical Analysis

In the next section, the results of the mechanical analysis are shown; namely, the change in fiber volume content, the related changes in the engineering properties, and the final deformation and residual stresses.

In Figure 18, the change in the fiber volume content is shown. Related to the initial degree of porosity (Equation (15)), the starting value of the fiber volume ratio varies for the considered samples. In the final application, the goal is to achieve a fiber volume content of 0.573, corresponding to the fully consolidated fiber filament. This was reached for all samples which were consolidated above melting point, irrespective of the pressure applied.

The process-dependent elastic moduli in the fiber and transverse direction are presented in Figure 19. Related to the changes in the fiber volume content, the initial and final stiffness also change during the process. Figure 19 shows the elastic moduli in fiber and transverse direction for sample A (fully consolidated in the end) as well as sample B (not fully consolidated in the end, approximately 4% porosity). As can be seen, the content of porosity influences the final stiffness as it is described in literature [9,20].

For sample B, the variation of temperature leads to a porosity variation, which also leads to an inhomogeneous distribution of the pressure as well as the engineering properties (deviation within the domain from the average value presented on Figure 19 is about 15%). Examples of inhomogeneous distributions for sample B are presented in Figure 20. For Samples A and C–E, the distribution of the final engineering properties is almost homogeneous due to the zero final porosity everywhere in the domain.

The final engineering properties are listed in Table 5 and are compared to the experimentally derived values from Table 1. Samples C and D have the same final engineering properties as sample A since all of the samples were fully consolidated with zero final porosity. Therefore, Table 5 presents resulting values only for samples A, B, and E

In the next section, the compaction and resulting thickness from the experimentally measured data are compared to the simulation results. Table 6 shows the comparison of the results. As can be seen, the resulting simulated thickness matches the simulated thickness quite well, with the average deviation being 1.2%.

Figure 21 presents the thickness change and the imposed strain in the thickness direction during the consolidation process for samples A and B. The strain is evaluated according to Hooke’s law. The developed mechanical approach corresponds to the elastic material, which limits the maximum consolidation strain, at which the model is robust.

Lastly, the directional stresses occurring during the additive fusion process are displayed for samples A and B in Figure 22. In the beginning of the consolidation stresses in fiber direction are positive due to the boundary conditions implemented (mold fixation) and negative after the crystallization due to the negative thermal expansion of the composite in the fiber direction (Figure 22a). In contrast, stresses in the transverse direction are positive during the cooling, which is caused by thermal shrinking (Figure 22b). The external pressure is applied in the thickness direction, therefore stresses in this direction are initially positive due to the thermal expansion and negative after the pressure application (Figure 22b). All the stresses are released after the demolding, which results in the near-zero final residual stresses due to the simple geometry of the considered specimen. This situation will change if the complex geometry is considered [20,25,26].

## 5. Conclusions

This paper presents an insight view into the consolidation of additive manufactured composite parts. These parts have a unique form of porosity because it is mainly based on the printing process in between the filaments and the porosity which has been determined by CT measurements to be around 30%. Due to this special type of porosity, a unique understanding of the void migration is needed to enable these parts to be used in structural application. 

The proposed finite-element model validated on experimental trails enables prediction of the final deformations, stresses, and void content as well as the internal void pressure and directional squeeze pressure depending on the consolidation setup considering transversal isotropic composite properties. The fiber volume ratio depends on porosity, and therefore the model respects local variations of mechanical properties.

In general, the model provides a good correspondence to the measured values of the final compaction (minimal model accuracy is 96%) and porosity (minimal model accuracy is 2%). Inaccuracies could be explained by multiple factors, such as the following:The assumptions made, such as the simplification of the porosity approach including the dimensionless approach, for the diffusion and stiffness dependency on the crystallization and porosity.Uncertainty in the initial porosity, sample linear dimensions and shape measurements, as well as the limitations of the CT-analysis.No consideration of the initial local porosity inhomogeneity due to the printing setup.

Model precision could be improved with consideration of the viscoelastic model for the matrix material and an advanced air bubble collapse model, as well as the consideration of local porosity distribution depending on the printing path and composite part assembly into the mold. 

The presented approach provides digital modeling of the consolidation process, decreasing the number of expensive prototyping iterations. The highly accurate 3D-printing and post-printing consolidation—together with the ANSYS finite-element model and fiber filament layup design—enables the transition from special applications to the serial production of additive manufactured continuous fiber composite parts.

In a future study, the authors intend to consider a complex composite part (see Figure 1) which consists of a mix of materials (PA12 and PA12-CF) and is consolidated with pressure application in multiple directions to prove that the developed model can be applied to real industrial problems.

## Figures and Tables

**Figure 1 polymers-14-03429-f001:**
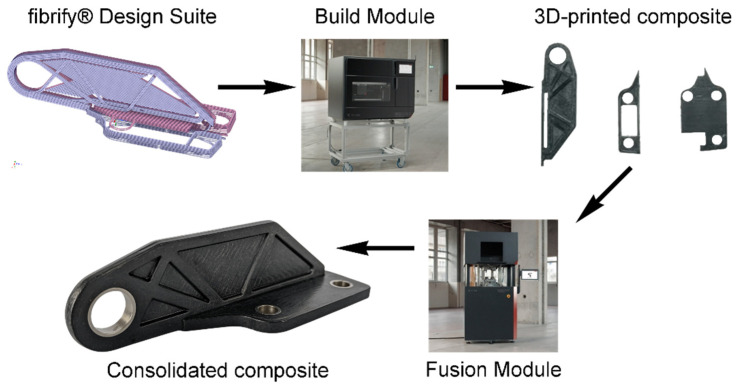
9T Labs’ production cycle of the composite part.

**Figure 2 polymers-14-03429-f002:**
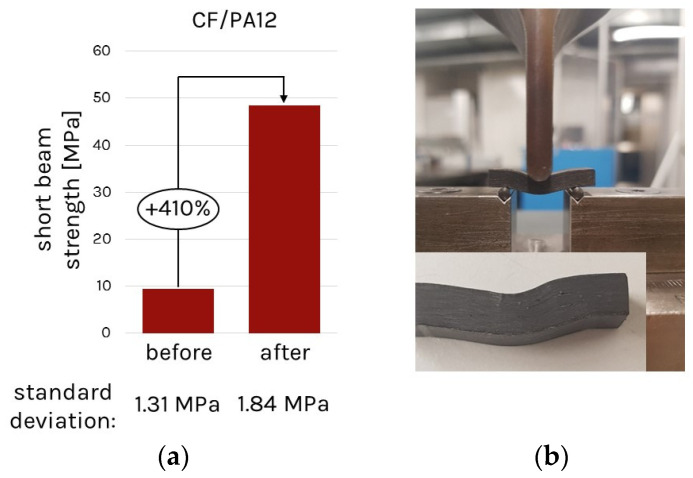
Preliminary study on the change in short beam strength for the PA12/CF material before and after fusion: (**a**) comparison of the strength before and after the consolidation; (**b**) experimental setup and resulting part’s deformations.

**Figure 3 polymers-14-03429-f003:**
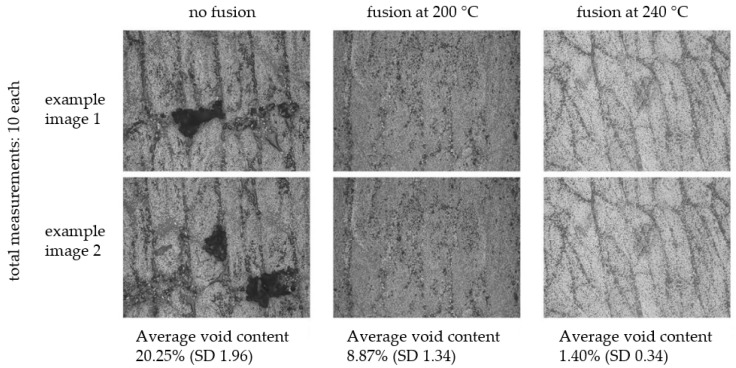
Preliminary study on change in the residual void content of the PA12/CF composite before and after fusion at different processing temperatures.

**Figure 4 polymers-14-03429-f004:**
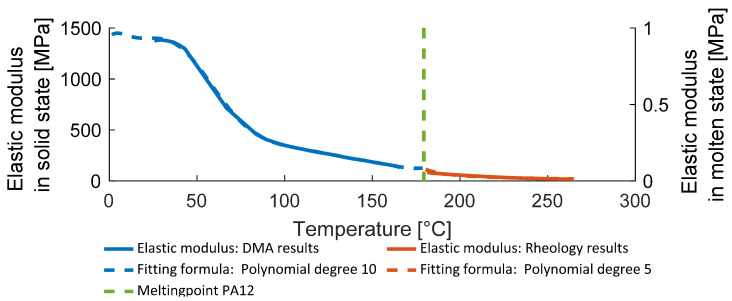
Measured and fit PA12 elastic modulus in solid and molten state.

**Figure 5 polymers-14-03429-f005:**
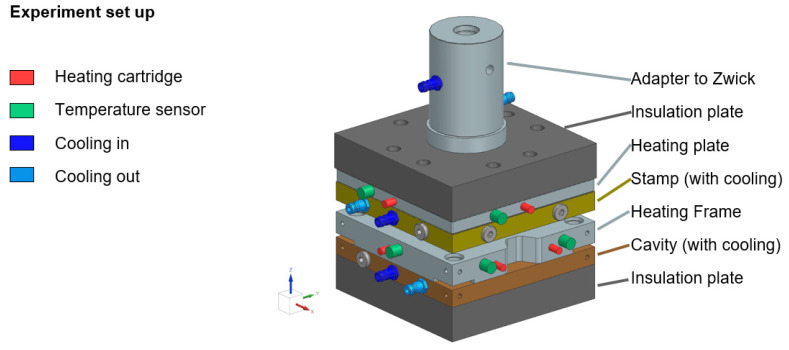
Consolidation experiment setup.

**Figure 6 polymers-14-03429-f006:**
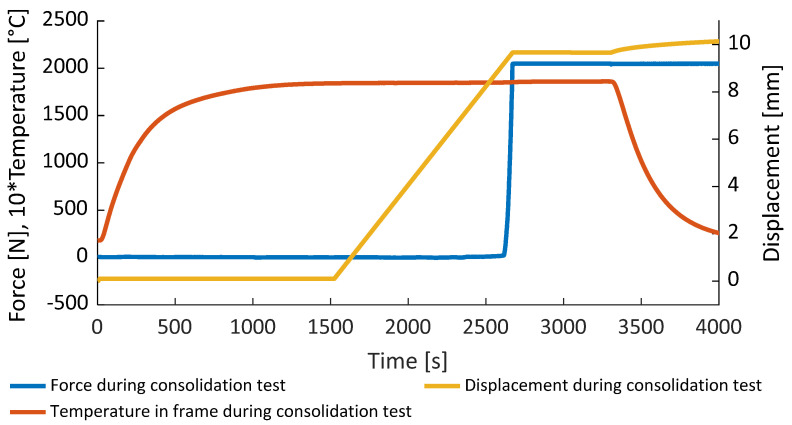
Force, temperature, and mold’s displacement measured during experiments.

**Figure 7 polymers-14-03429-f007:**
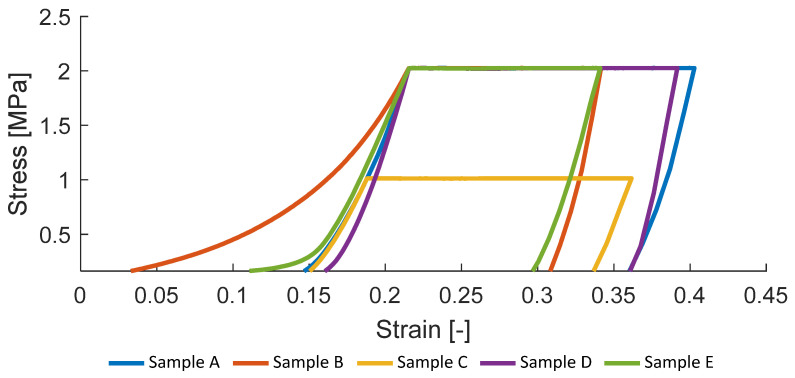
Measured stresses depending on strain for samples A–E.

**Figure 8 polymers-14-03429-f008:**
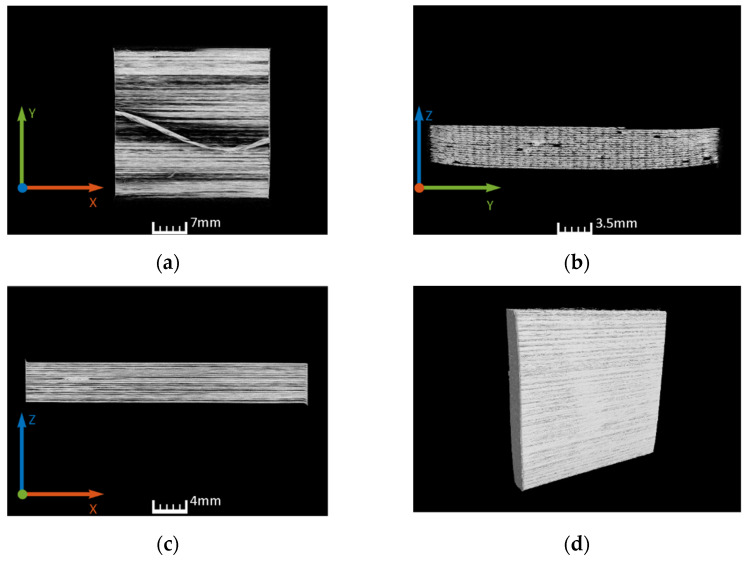
CT scan of the unconsolidated blockply specimen: (**a**) xy-plane crossection; (**b**) xz-plane crossection; (**c**) xz-plane crossection; (**d**) 3D overview.

**Figure 9 polymers-14-03429-f009:**
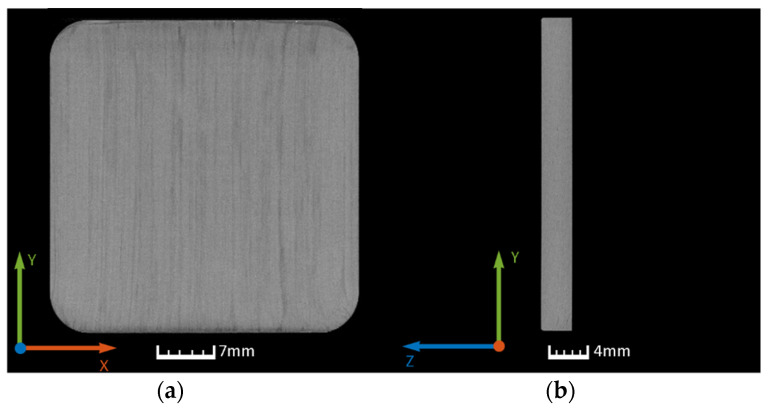
CT scan of the consolidated blockply specimen A: (**a**) xy-plane crossection; (**b**) yz-plane crossection.

**Figure 10 polymers-14-03429-f010:**
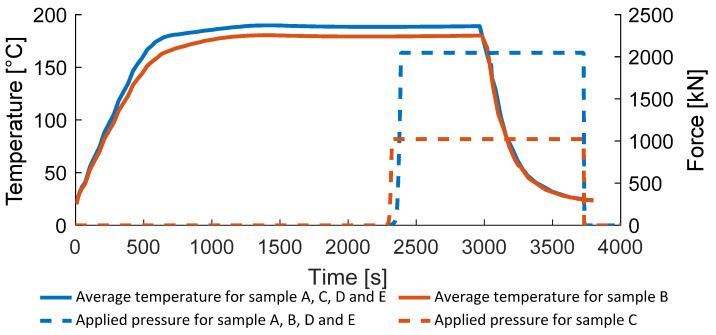
Temperature and pressure applied in the experiments.

**Figure 11 polymers-14-03429-f011:**
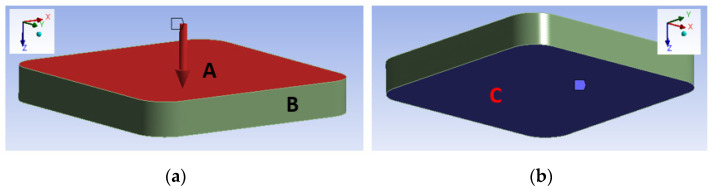
Boundary conditions for the mechanical problem: (**a**) the pressure is applied to surface A (depicted in red); surfaces A and B are fixed in the x and y direction; (**b**) surface C is fixed in the z direction (depicted in blue).

**Figure 12 polymers-14-03429-f012:**
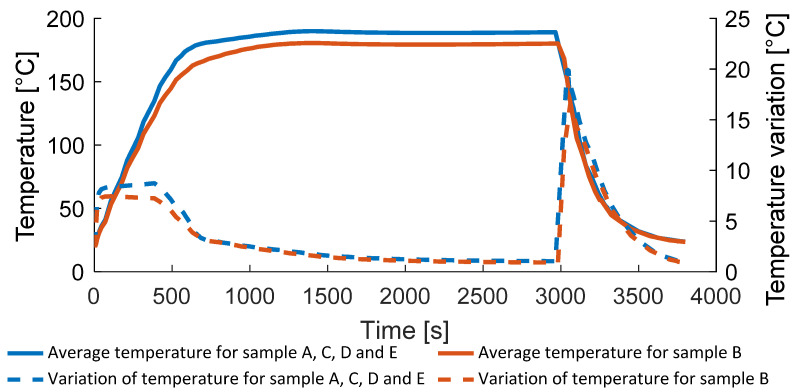
The average domain temperature and temperature variation.

**Figure 13 polymers-14-03429-f013:**
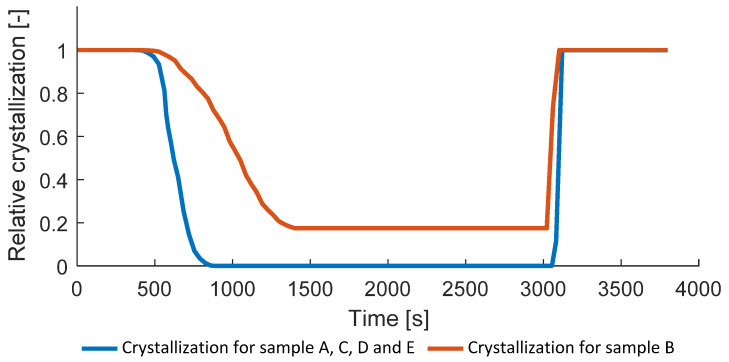
Relative degree of crystallization.

**Figure 14 polymers-14-03429-f014:**
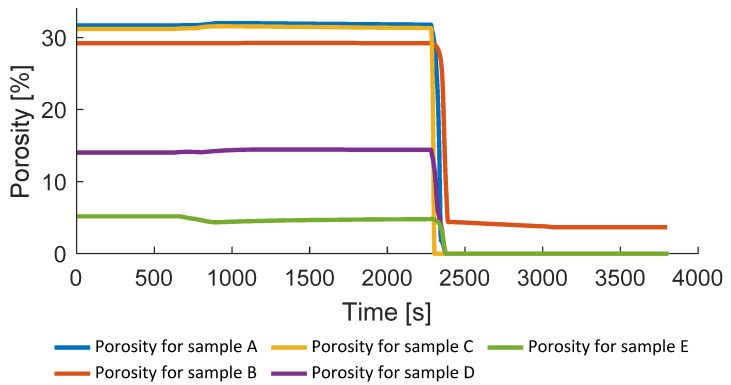
Porosity evolution during the consolidation.

**Figure 15 polymers-14-03429-f015:**
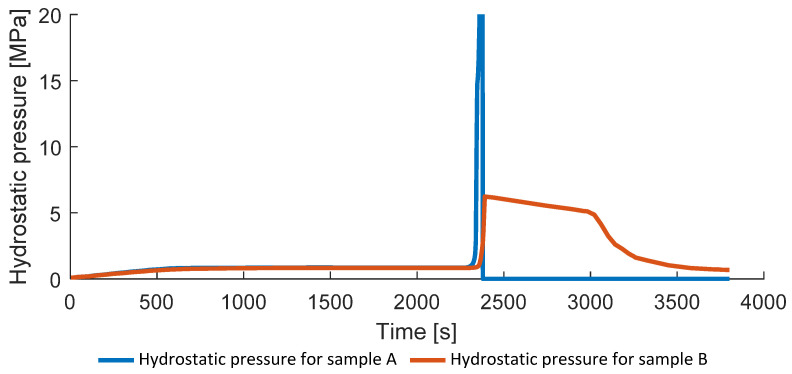
Hydrostatic void pressure.

**Figure 16 polymers-14-03429-f016:**
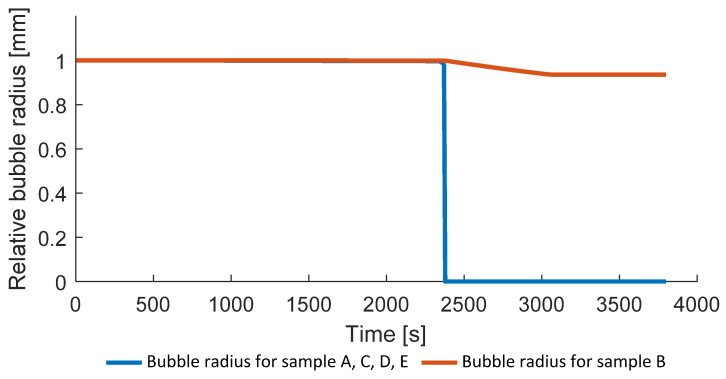
Relative bubble radii.

**Figure 17 polymers-14-03429-f017:**
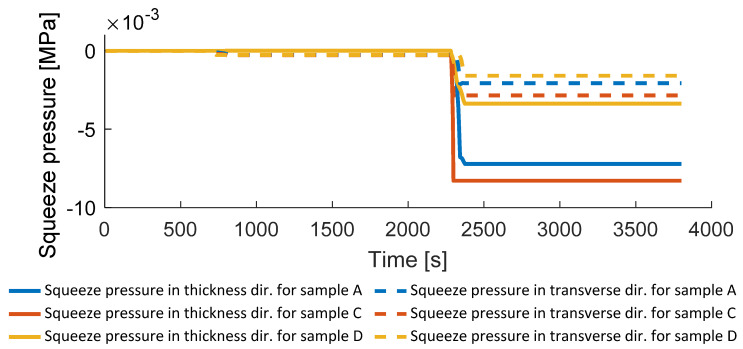
Squeeze pressures transverse to fiber and thickness directions.

**Figure 18 polymers-14-03429-f018:**
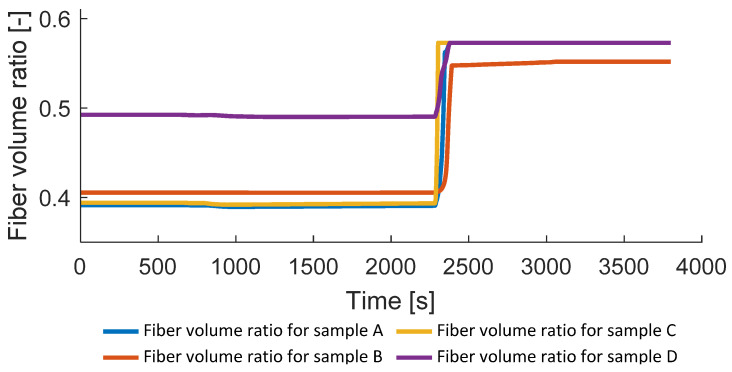
Fiber volume ratio.

**Figure 19 polymers-14-03429-f019:**
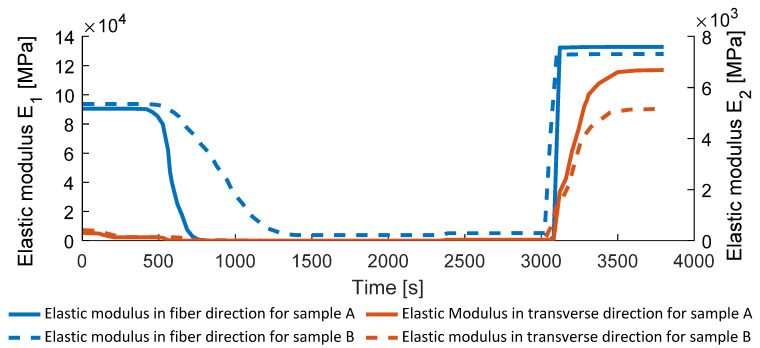
Elastic modulus.

**Figure 20 polymers-14-03429-f020:**
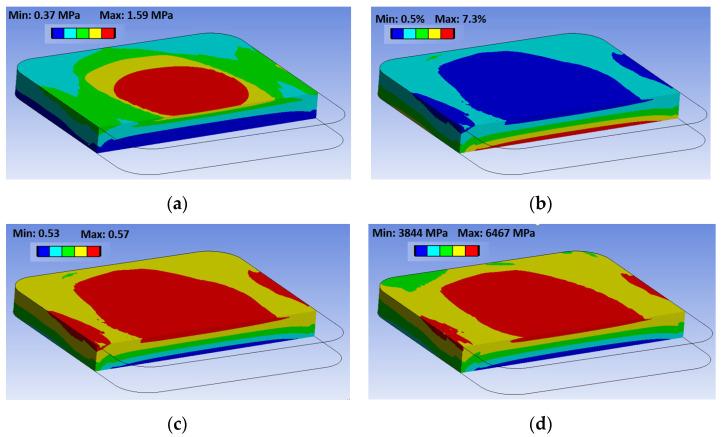
Final void pressure (**a**), porosity (**b**), fiber volume content (**c**), and elastic modulus in the transverse to fiber direction (**d**) in sample B (2 MPa, 175 °C).

**Figure 21 polymers-14-03429-f021:**
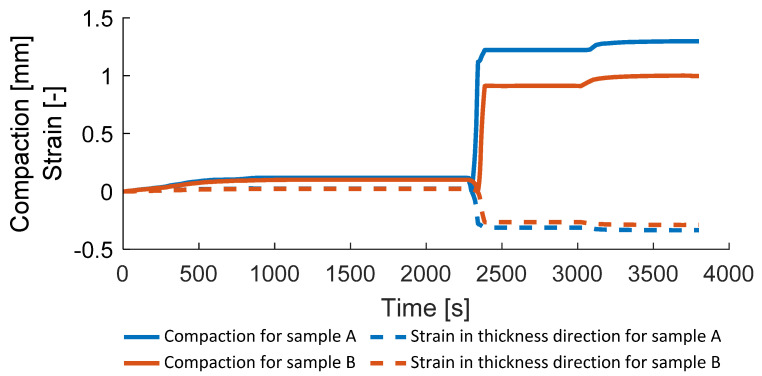
Compaction and strain of samples A and B.

**Figure 22 polymers-14-03429-f022:**
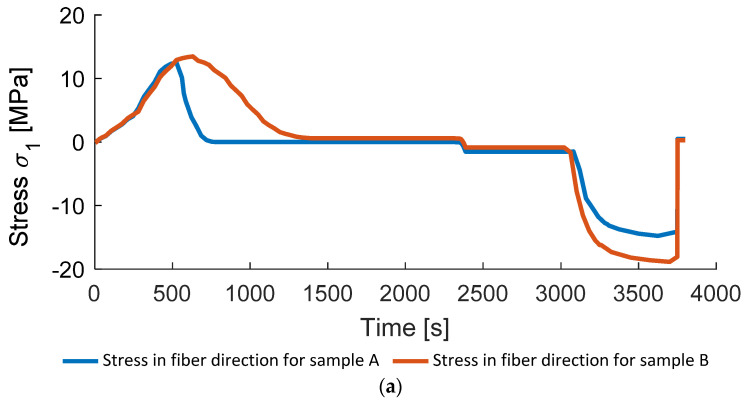
Directional stresses in samples A and B: (**a**) in fiber direction; (**b**) transverse to fiber direction; (**c**) in thickness direction.

**Table 1 polymers-14-03429-t001:** Material properties at room temperature.

Material	Property	Identifier and Value	Unit
PA12	Young’s modulus	E_m_ = 1372	MPa
PA12	Poisson’s ratio	ν_m_ = 0.43	-
PA12	Coefficient of thermal expansion (CTE)	α_m_ = 136.83 × 10^−6^	K^−1^
PA12	Glass transition temperature	T_g_ = 94	°C
PA12	Melting temperature	T_m_ = 179	°C
PA12	Crystallization shrinkage	3.0	%
Fiber	Young’s modulus in fiber direction	E_f,1_ = 231,000	MPa
Fiber	Young’s modulus transverse to fiber direction	E_f,2_ = 28,000	MPa
Fiber	Shear modulus	G_f,12_ = 28,600	MPa
Fiber	Poisson’s ratio	ν_f_ = 0.23	-
Fiber	CTE in fiber direction	α_f,1_ = −2.23 × 10^−6^	K^−1^
Fiber	CTE transverse to fiber direction	α_f,2_ = 15.7 × 10^−6^	K^−1^
PA12-CF	Young’s modulus in fiber direction	E_1_ = 132,632	MPa
PA12-CF	Young’s modulus transverse to fiber direction	E_2_ = 6568	MPa
PA12-CF	Shear modulus	G_12_ = 2035	MPa
PA12-CF	Poisson’s ratio	ν_12_ = 0.298	-
PA12-CF	CTE in fiber direction	α_1_ = −1.609 × 10^−6^	K^−1^
PA12-CF	CTE transverse to fiber direction	α_2_ = 67.417 × 10^−6^	K^−1^
PA12-CF	CTE in out of plane direction	α_3_ = 90.257 × 10^-6^	K^−1^
PA12-CF	Fiber volume content (fully consolidated)	φ_0_ = 0.573	-

**Table 2 polymers-14-03429-t002:** Printing parameters.

Material	Parameter	Value	Unit
PA12	Nozzle temperature	230	°C
PA12	Heat bed temperature	90	°C
PA12	Layer height	0.18	mm
PA12	Layer width	0.65	mm
PA12-CF	Nozzle temperature	220	°C
PA12-CF	Heat bed temperature	90	°C
PA12-CF	Layer height	0.18	mm
PA12-CF	Layer width	0.8	mm

**Table 3 polymers-14-03429-t003:** Considered blockply samples and their consolidation setups.

Sample	Consolidation Pressure, MPa	Consolidation Temperature, °C	Initial (Final) Thickness, mm	Initial (Final) Porosity, %
A (PA12-CF)	2	185	4.54 (3.09)	31.67 (0.05)
B (PA12-CF)	2	175	4.42 (3.35)	29.22 (4.96)
C (PA12-CF)	1	185	4.55 (3.07)	31.19 (0.07)
D (PA12-CF)	2	185	4.11 (3.52)	14.04 (0.0)
E (PA12)	2	185	4.44 (4.18)	5.18 (0.0)

**Table 4 polymers-14-03429-t004:** Measured and simulated final porosity.

Sample	Initial Porosity, %	Final Porosity, %	Simulated Final Porosity, %
A (PA12-CF)	31.67	0.04	0.0
B (PA12-CF)	29.22	4.96	3.7
C (PA12-CF)	31.19	0.07	0.0
D (PA12-CF)	14.04	0.0	0.0
E (PA12)	5.18	0.0	0.0

**Table 5 polymers-14-03429-t005:** Final engineering properties’ validation.

Sample	Material Property, Unit	Initial Model Value	Final Model Value	Measured Value for the Fully Consolidated Material
A (PA12-CF)	E_1_, MPa	90,493	132,860	132,632
A (PA12-CF)	E_2_, MPa	299	6684	6568
A (PA12-CF)	G_12_, MPa	76	2076	2035
A (PA12-CF)	ν_12_, -	0.35	0.31	0.298
A (PA12-CF)	φ, -	0.39	0.573	0.573
B (PA12-CF)	E_1_, MPa	93,749	127,970	132,632
B (PA12-CF)	E_2_, MPa	405	5156	6568
B (PA12-CF)	G_12_, MPa	103	1514	2035
B (PA12-CF)	ν_12_, -	0.34	0.31	0.298
B (PA12-CF)	φ, -	0.40	0.55	0.573
E (PA12)	E_m_, MPa	942	1400	1372
E (PA12)	G_m_, MPa	329	489	479
E (PA12)	ν_m_	0.43	0.43	0.43

**Table 6 polymers-14-03429-t006:** Measured and simulated final sample compaction.

Sample	Experimental Compaction, mm	Simulated Compaction, mm	Compaction Error Relative to the Initial Thickness, %
A (PA12-CF)	1.46	1.3	3.6
B (PA12-CF)	1.07	1.01	1.3
C (PA12-CF)	1.48	1.48	0.0
D (PA12-CF)	0.59	0.58	0.2
E (PA12)	0.26	0.22	0.9

## Data Availability

The data presented in this study are available upon request from the corresponding author.

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
