# Peer review of "Consolidation of Additive Manufactured Continuous Carbon Fiber Reinforced Polyamide 12 Composites and the Development of Process-Related Numerical Simulation Methods"

_polymers, 2022, doi:10.3390/polym14163429_

Round 1

Reviewer 1 Report

Title:

Needs to be more precise, it is very generic in terms of additive manufacturing research literature

Abstract:

Does not completely encapsulate the research presented. It should flow as follows: quick introduction; problem; brief info on status quo; the problem that is being solved by this research; brief info about results. Kindly rephrase it to reflect the same. 

Introduction:

Very scattered along many different ideas. 

Need to stay focussed, briefly introduce the topic and move towards relevant literature, define the literature gaps specifically, preferably numbered in bullet points, then move towards the proposed solution, and finally, clear and concise objectives/aims of the research should be stated (again in bullet points), the final paragraph should give a brief outline of the rest of the sections including results. 

Also, the literature summary, gaps, and objectives need to be clearly separated, which does not seem to be the case here. if this is a continuation of a research article or commercial research (as it seems here), please specify it objectively and give proper references. 

Section 2 and 3

The problem persists here as well. Materials and methods and the development of equations include a lot of referred equations from the literature and hardly any new equations or empirical relations are developed (or there are but are not evident). Please differentiate between the literature and this section clearly. Materials and methods also include a lot of result plots and charts, which should be moved to results section. 

Results: 

Explicitly explain results, compare with literature (section 1), discuss, outline novel findings, limitations

Conclusions:

Give overall picture of the research status quo, presented solution, brief results, limitations leading to future work

Author Response

Reply to Reviewer 1 comments:

  1. The title is modified to provide better specification of the paper content.
  2. The abstract is updated according to the proposed structure.
  3. The introduction is modified highlighting points proposed by the Reviewer.
    The literature gaps are presented via bullet points (lines 93-117).
    Definition of the paper objectives is added to the introduction (lines 142-155), while a brief description of the results already concluded the introduction (lines 156-166).
  4. Constitutive equations development section indeed includes a lot of referred equations used for the mathematical model development. The referred equations are not new; however, it is essential to provide all the formulas allowing to define all the engineering properties of the composite, otherwise the study could not be reproduced and verified by other researchers. For example, formulas (21) and (24) are well-known Halpin-Tsai equations, which contain model parameters (those parameters are now highlighted, lines 393-395 and 397-399) needed to be fit to the experimental data for a specific material (in our case PA12-CF). Without the representation of these formulas including all the model constants and parameters it would be not possible to develop the material model. Additionally, referred equations combination and implementation to the model are novel, allowing to precisely describe the mechanical properties dependency on the fiber volume ratio for PA12-CF composite used in the presented study.
    Developed by the authors empirical relations are now marked in the main text (lines 266-267, 332-334, 370-372, 386-388).
    Section 2 Materials and Methods title was changed to Materials, Methods and Characterization to provide better description of the section content. Results of this section (like Elastic modulus dependency on temperature, porosity evaluation and Bulk modulus evaluation) are used for the model development in section 3 Development of Constitutive Equations, while section 4 Model Application and Validation presents the simulation results. Therefore, results of the material characterization are presented in section 2 and not in section 4.
  5. Results are explained in more detail including the discussion of the findings and the limitations of the proposed approach and the comparison with literature (lines 479-482, 495-498, 500-511, 520-523, 553, 576-578, 593-602).
  6. Conclusion is updated according to the structure proposed by the Reviewer.

Reviewer 2 Report

This manuscript reports a consolidation process study of additive manufactured continuous fiber composite parts by experiment, theory and simulation. The proposed simulation workflow has a good practical value in the field of composite additive manufacturing, especially for decreasing the number of expensive prototyping iterations. Questions and comments to address are the following:

- The most significant advantage of 3D-printing technology is to manufacturing structures with complex geometry. Therefore, a relative complex sample should be added in the manuscript to prove the versatility and feasibility of the proposed simulation method.

- In the simulation part, it is weird that only some sample results are illustrated in Figs. 15, 17, 18, 19, 21, 22. In addition, mechanical characterizations of samples are too rough.

- What types of strain are used in Figs. 21 and 22? What is the strain definition?

- The authors should check the tables 1 and 5 carefully. Some symbols and formats are not uniform.

- The parameter φ in Eq. (10) is not defined.

Author Response

Reply to Reviewer 2 comments:

  1. The goal of the presented study is to describe the developed porosity approach, demonstrate its application to the set of simple cases, tune and validate the proposed model. Considered “blockply” samples have simple shape, which allows better understanding of the main issues of the proposed approach as well as the mathematical model check and validation by other researchers.
    However, we totally agree with the Reviewer`s comment, that the developed approach feasibility must be shown on the composite part with complex shape. The solution for the complex geometry was presented on ECCM20 (conference proceedings will be available in Autumn 2022), where the complex composite part (a bracket to assemble the helicopter door) from the submitted manuscript`s Figure 1 is considered.
    In the Conclusion section it is mentioned (lines 690-693) that consolidation of the part with complex geometry will be considered and described in detail in the future work.
  2. We want to avoid a mess in the plots providing less data for the better readability. Therefore, generally the two most differently consolidated samples A and B were discussed in detail since the solution for other samples is almost identical to one of the samples A or B due to the similarity of the consolidation setup. If solutions are significantly different for all the samples – they all are presented on the plot. Solution for all the samples is shown for the compaction (Table 6), porosity (Figure 14), squeeze pressure (Figure 17, except samples B where no fully molten state was reached and sample E where no composite material is considered) and fiber volume ratio (Figure 18, except sample E where no composite material is considered).
    Figure 15: All the samples except B show the same behavior as sample A since the pressure and temperature applied in the experiment are the same or do not influence much the hydrostatic pressure.
    Figure 17: Samples A, C and D are presented. There is no squeeze flow for sample B (since no full melting occurred) and sample E (since the material there is pure matrix PA12).
    Figure 18: All the samples except E are presented on the plot, sample E is pure PA12 and therefore the fiber volume ratio is always 0.
    Figure 19: All the samples except B has zero final porosity and, therefore, their Elastic moduli evolution during the process are almost identical to the sample A.
    Figure 21: Compaction rate and strain are indeed a bit different for all the samples. However, the trend is always the same. Therefore, only final values of compaction are shown in the Table 6, while the time series are not shown on the plot for better readiness.
    Figure 22: Same as Figure 21, only samples A and B are shown since other samples are fully molten during the consolidation and, therefore, demonstrate the similar to sample A stress time series.
    Corresponding comments explaining missing results are added to the main text.
    Presented mechanical characteristics of samples include all the engineering properties required for the definition of the Jacobian. All mentioned properties are listed on the Table 1 and based on the extensive characterization. Values of other material properties or characterization details can be provided based on request.
  3. Presented strain is the engineering strain, in Ansys it is calculated via Hooke`s law. The presented values correspond to the normal strain in thickness direction. This information is added to the manuscript text (lines 576-578).
  4. Format and symbols in Tables 1 and 5 are fixed to be uniform in the manuscript.
  5. The parameter φ in Eq. (10) is now defined (line 350).

Round 2

Reviewer 1 Report

Thanks for doing the revisions. 

Reviewer 2 Report

The authors have addressed all my comments. The manuscript is now ready for publication.